# Elimination of paternal mitochondria in mouse embryos occurs through autophagic degradation dependent on PARKIN and MUL1

Rebecca Rojansky, Moon-Yong Cha, David C Chan*

Division of Biology and Biological Engineering, California Institute of Technology, Pasadena, United States

**Abstract** A defining feature of mitochondria is their maternal mode of inheritance. However, little is understood about the cellular mechanism through which paternal mitochondria, delivered from sperm, are eliminated from early mammalian embryos. Autophagy has been implicated in nematodes, but whether this mechanism is conserved in mammals has been disputed. Here, we show that cultured mouse fibroblasts and pre-implantation embryos use a common pathway for elimination of mitochondria. Both situations utilize mitophagy, in which mitochondria are sequestered by autophagosomes and delivered to lysosomes for degradation. The E3 ubiquitin ligases PARKIN and MUL1 play redundant roles in elimination of paternal mitochondria. The process is associated with depolarization of paternal mitochondria and additionally requires the mitochondrial outer membrane protein FIS1, the autophagy adaptor P62, and PINK1 kinase. Our results indicate that strict maternal transmission of mitochondria relies on mitophagy and uncover a collaboration between MUL1 and PARKIN in this process.

*For correspondence: dchan@caltech.edu

Competing interests: The authors declare that no competing interests exist.

## Introduction

In most animals, including mammals, mitochondria are inherited strictly through the maternal lineage. Because sperm deliver mitochondria into the egg during fertilization, mechanisms likely exist to eliminate paternal mitochondria from the early embryo. Uniparental inheritance of mitochondria ensures that only one haplotype of mitochondrial DNA (mtDNA) exists in the offspring, a phenomenon with considerable biomedical implications. It underlies the maternal inheritance of diseases caused by mutations in mtDNA (*Carelli and Chan, 2014*) and enables the use of mtDNA sequences to track human migrations during evolution. Mouse studies suggest that extensive heteroplasmy, the co-existence of more than one haplotype of mtDNA, is genetically unstable and associated with physiological abnormalities (*Sharpley et al., 2012*).

Although uniparental inheritance is a defining characteristic of mitochondria, there is much speculation about its mechanism in vertebrates (*Carelli, 2015*). Most of our knowledge has come from invertebrate model organisms. The phenomenon has been most decisively dissected in *Caenorhabditis elegans*, where paternal mitochondria are eliminated by mitophagy (*Al Rawi et al., 2011*; *Sato and Sato, 2011*; *Zhou et al., 2011*), a process in which mitochondria are engulfed by autophagosomes and delivered to lysosomes for destruction. In *Drosophila melanogaster*, paternal mitochondrial elimination involves autophagic components but occurs independently of PARKIN (*Politi et al., 2014*), a Parkinson's disease-related E3 ubiquitin ligase that is central to the most heavily studied mitophagy pathway (*Pickrell and Youle, 2015*). However, it is unclear to what extent these insights from invertebrate model organisms extend to mammals. Consistent with a role for

**eLife digest** Mitochondria are commonly referred to as the 'powerhouses' of animal cells because these structures provide the majority of the energy in most cells. People inherit their mitochondria from their mother, and not their father. This is because the father's mitochondria, which are delivered by sperm to the egg, are degraded early on when the embryo starts to develop.

Previous studies with model organisms, like nematode worms, showed that mitochondria delivered via sperm (also known as 'paternal mitochondria') were delivered to structures called lysosomes and broken down by the enzymes contained within. However, it remained controversial whether this process, named mitophagy, also occurred in mammalian cells, and the molecules involved were unknown.

Now, Rojansky et al. have identified key molecules that are essential for the degradation of mitochondria in mouse cells and show that these same molecules are needed to degrade paternal mitochondria in early mouse embryos. These results indicate that paternal mitochondria are indeed degraded by mitophagy in mice. In addition, Rojansky et al. also note that one of the key molecules is a protein called PARKIN, which is mutated in many inherited cases of Parkinson's disease, a major neurodegenerative disorder.

Even though these new findings provide a clearer idea as to *how* paternal mitochondria are degraded, the question of *why* remains unanswered. As a result, it is likely that this topic will continue to be heavily debated. Nevertheless, having identified the key molecules involved in degrading paternal mitochondria, it may now be possible to address this question more directly – for example by interfering with this process and then examining the consequences.

autophagy, sperm mitochondria from mice are ubiquitinated (*Sutovsky et al., 1999*) and, after fertilization, are immuno-positive for P62 and the ATG8 homologs LC3 and GABARAP (*Al Rawi et al., 2011*). However, a subsequent study in mouse disputed the role of autophagy in elimination of paternal mitochondria (*Luo et al., 2013*). The association of LC3 with paternal mitochondria was observed to be transient and occurred well before paternal mitochondrial elimination. In addition, it was found that paternal mitochondria were segregated unevenly to blastomeres during early embryonic cell division. Based on these results, the authors rejected the role of autophagy and advocated a passive dilution mechanism whereby murine paternal mitochondria are stochastically lost due to uneven segregation to the cells of the embryo (*Luo et al., 2013*).

This mechanistic uncertainty highlights the need to move beyond correlative studies relying on co-localization of autophagy markers with paternal mitochondria, and instead to perform functional studies that directly test the role of autophagy. In *C. elegans*, the functional role of autophagy was revealed by the persistence of paternal mitochondria in embryos depleted for core autophagy genes, such as the ATG8 homologs LGG-1 and LGG-2 (*Al Rawi et al., 2011*; *Sato and Sato, 2011*; *Zhou et al., 2011*). A similar approach is not feasible in mouse, however, because disruption of basal autophagy results in embryonic arrest at the four-cell stage (*Tsukamoto et al., 2008*), well before paternal mitochondria are normally eliminated.

To circumvent this technical hurdle, we reasoned that a functional test for the role of mitophagy might be possible by focusing on mitophagy-specific genes, whose depletion would be less likely to arrest early embryonic development compared to core autophagy genes. To obtain a set of candidate mitophagy genes, we first characterized the requirements for mitophagy in cultured cells. These experiments led to the realization that two E3 ubiquitin ligases, PARKIN and MUL1, synergistically function in degradation of mitochondria. We then used a gene disruption approach in early embryos to show that mitophagy mediates the degradation of paternal mitochondria.

## Results

### A functional assay for elimination of paternal mitochondria

To develop an assay to track paternal mitochondria in the early mouse embryo, we utilized male *PhAM* mice, in which all mitochondria, including those in the sperm midpiece, are labeled with a

mitochondrially-targeted version of the photoconvertible Dendra2 fluorescent protein (*Pham et al., 2012*) (*Figure 1A*). When male *PhAM* mice were mated with wild-type females, the resulting embryos contained brightly fluorescent paternal mitochondria. At 12 hr post-fertilization (*Figure 1B*), the paternal mitochondria were found in a linear cluster, reflecting their original, compact organization in the sperm midpiece. At 36 hr after fertilization (*Figure 1C*), this cluster began to disperse in cultured embryos, and thereafter, well-separated individual mitochondria were visible within blastomeres. Over the next 2 days, paternal mitochondrial content progressively decreased (*Figure 1D–F*). At 84 hr after fertilization, the majority of embryos had lost all paternal mitochondria (*Figure 1F*). Quantification of these results showed a reproducible and progressive loss of paternal mitochondria between 60 and 84 hr post-fertilization (*Figure 1G*). To determine whether this pattern is specific to paternal mitochondria, we additionally mated *PhAM* female mice with wild-type males, resulting in embryos with fluorescent maternal mitochondria. In these embryos, there was no reduction in the maternal mitochondrial content between 60 and 84 hr post-fertilization (*Figure 1H*, *Figure 1—figure supplement 1*).

We used a lentiviral approach to functionally probe the role of autophagy genes in this process (*Figure 1I*). We microinjected one-cell stage zygotes with lentivirus encoding mCherry and control shRNA or shRNA targeting the core autophagy gene *Atg3*. In embryos injected with lentivirus, the mCherry reporter was expressed within 48 hr of injection (60 hr post-fertilization). When nontargeting shRNA was expressed, development of the embryo was unaffected, and Dendra2-positive mitochondria were eliminated by 84 hr with the usual kinetics (*Figure 1J*). In embryos injected with shRNA against *Atg3* (*Figure 1K*), however, embryo development was arrested at the four-cell stage, consistent with a previous report using *Atg5*-null oocytes (*Tsukamoto et al., 2008*). Similarly, treatment of embryos with bafilomycin, an autophagy inhibitor, arrested embryonic development (*Figure 1L*). In both cases, the treated embryos showed persistence of paternal mitochondria at 84 hr. However, due to the early disruption of embryonic development, it was not possible to conclude if autophagy has a specific role in elimination of paternal mitochondria. This result indicated that disruption of core autophagy genes in this system is not a viable experimental approach. We therefore decided to focus on mitophagy-specific genes. We injected embryos with lentivirus encoding shRNA against *Parkin* (*Park2*), an E3 ubiquitin ligase that is central to the most studied pathway for mitopahgy (*Durcan and Fon, 2015*; *Pickrell and Youle, 2015*). Such embryos show loss of paternal mitochondria by 84 hr after fertilization, suggesting that the process occurs in the absence of PARKIN (*Figure 1M*).

## FIS1, TBC1D15, and P62 are essential for OXPHOS-induced mitophagy in MEFs

Given the negative results with PARKIN, we turned to cultured cells, where the role of specific proteins in mitophagy could be more readily analyzed. Our strategy was to identify, in cultured cells, a small set of mitophagy genes, which could then be re-analyzed in early embryos. To monitor mitophagy, we constructed a dual color fluorescence-quenching assay based on an EGFP-mCherry reporter localized to the mitochondrial matrix. Normal mitochondria are yellow, having both green and red fluorescence in the matrix, whereas mitochondria within acidic compartments show red-only fluorescence, due to the selective sensitivity of EGFP fluorescence to low pH. A similar approach using a mitochondrial outer membrane EGFP-mCherry reporter has been effective for monitoring mitophagy (*Allen et al., 2013*). When mouse embryonic fibroblasts (MEFs) were cultured with a moderate concentration (10 mM) of glucose, a condition in which their metabolism relies largely on glycolysis, they showed few red-only mitochondria (*Figure 2A*). We previously defined a glucose-free, acetoacetate-containing culture formulation that induces MEFs to substantially upregulate OXPHOS activity (*Mishra et al., 2014*). When cells were cultured for 4 days in this OXPHOS-inducing medium, many cells exhibited numerous red puncta (*Figure 2A*). This observation is consistent with a study showing that glucose-free conditions promote increased turnover of mitochondria (*Melser et al., 2013*) and likely reflects the higher turnover of mitochondria when the activity of the respiratory chain is elevated. *Atg3* knockout MEFs did not form red puncta under the OXPHOS-inducing condition (*Figure 2B–C*), indicating that formation of red puncta is dependent on the core autophagy machinery. Consistent with this idea, the level of lipidated LC3, another core component of the autophagy pathway, was elevated (*Figure 2D*). Moreover, the red-only puncta co-localized extensively with mTurquoise2-LC3B, suggesting that they represent mitochondrial contents within the

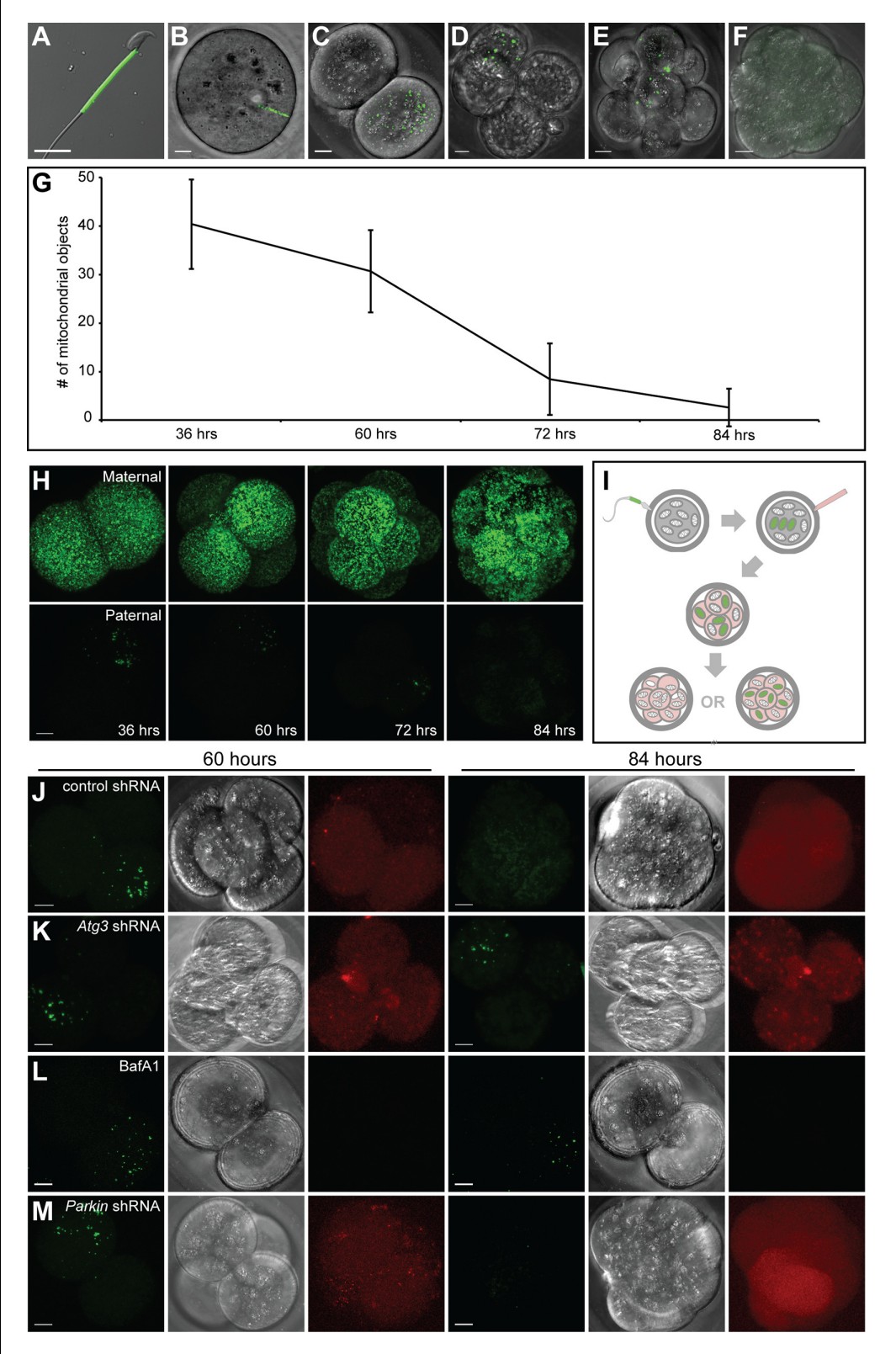

**Figure 1.** Paternal mitochondria are degraded by 84 hr after fertilization. (**A**) Fluorescence of mito-Dendra2 in a live sperm cell isolated from the cauda epididymis of a *PhAM* mouse. (**B–F**) Mito-Dendra2 in a 12 hr (**B**), 36 hr (**C**), 60 hr (**D**), 72 hr (**E**), and 84 hr embryo (**F**). In (**B**), note that mito-Dendra2 is circumscribed to a distinct rod-like structure. The mitochondria disperse in later embryos and are lost by 84 hr. (**G**) Quantification of the mito-Dendra2

*Figure 1 continued on next page*

*Figure 1 continued*

signal (see Materials and methods) at 36, 60, 72, and 84 hr after fertilization. Each data point represents the mean of 15 embryos. Error bars indicate SD. (**H**) Representative maximum intensity projection images of maternal mitochondrial content versus paternal mitochondrial content over time. Embryos with mito-Dendra2-labeled maternal mitochondria were derived from crosses of wildtype males with homozygous *PhAM* females. Embryos with labeled paternal mitochondria were derived from crosses of wild-type females with homozygous *PhAM* males, whose sperm donate Dendra2-labeled mitochondria to the embryo upon fertilization. Embryos were cultured in vitro and imaged at the indicated time. Note that paternal Dendra2 signal decreases with time, whereas maternal Dendra2 signal does not. (**I**) Schematic of paternal mitochondrial elimination assay. Wildtype females are mated with *PhAM* males. One-cell embryos are microinjected in the perivitelline space with concentrated lentivirus targeting candidate genes. During in vitro culture, embryos are periodically imaged live and monitored for their ability to eliminate paternal mitochondria. (**J**) Representative images of embryos injected with lentivirus carrying nontargeting shRNA. The left three images show mito-Dendra2, phase-contrast, and mCherry signals at 60 hr; the right three images show the same as 84 hr. (**K**) Embryos injected with lentivirus carrying *Atg3* shRNA. (**L**) Embryos treated with bafilomycin A1. (**M**) Embryos injected with lentivirus carrying *Parkin* shRNA. All scale bars are 10 μm.

The following figure supplement is available for figure 1:

**Figure supplement 1.** Persistence of maternal versus paternal mitochondria after fertilization.

autophagosome pathway (*Figure 2E*, arrows). In addition, a subset of the red puncta co-localize with LAMP1, likely indicating later intermediates that have progressed to lysosomes (*Figure 2F*). In contrast, in glycolytic medium, mTurquoise2-LC3B did not co-localize with mitochondria (*Figure 2E*). In addition, we found that p62 (SQSTM1), a protein implicated in autophagy (*Pankiv et al., 2007*) and mitophagy (*Seibenhener et al., 2013*), localized to mitochondria only under the OXPHOS-inducing condition (*Figure 2G*). Unlike LC3B and LAMP1, however, P62 was localized to both red punctate mitochondria and elongated yellow mitochondria. These results indicate that the OXPHOS-inducing condition results in an increase in mitophagy intermediates.

With this cellular system, we sought to identify genes required for induced mitophagy. Previous studies suggested that mitochondrial dynamics, particularly mitochondrial fission, is important for efficient mitophagy (*Mao et al., 2013*; *Tanaka et al., 2010*). To explore this idea, we examined the efficiency of OXPHOS-induced mitophagy in a panel of MEF cell lines deficient in mitochondrial fusion or fission genes: *Mitofusin 1* (*Mfn1*), *Mitofusin 2* (*Mfn2*), both *Mfn1* and *Mfn2* (*Mfn*-dm), *Optic atrophy 1* (*Opa1*), *Mitochondrial fission factor* (*Mff*), *Dynamin-related protein 1* (*Drp1*), and *Mitochondrial fission 1* (*Fis1*) (*Figure 3A*). MEFs deficient in mitochondrial fusion were competent for mitophagy. In fact, *Mfn*-dm cells and *Opa1-/-* cells showed substantial mitophagy even under glycolytic culture conditions, consistent with the findings that mitochondrial fusion protects against mitophagy (*Gomes et al., 2011*; *Rambold et al., 2011*) and that *Mfn*-dm cells have constitutive localization of Parkin to mitochondria (*Narendra et al., 2008*). Among cell lines deficient in mitochondrial fission, *Drp1-/-* and *Mff-/-* cells showed normal levels of mitophagy under OXPHOS conditions (*Figure 3A*).

In contrast, *Fis1-/-* cells had dramatically reduced mitophagy under OXPHOS conditions (*Figure 3A–B*), and a failure of both LC3 and P62 to co-localize with mitochondria (*Figure 3C–D*). Although FIS1 is a central player in yeast mitochondrial fission, it does not play a prominent role in mammalian mitochondrial fission (*Losón et al., 2013*; *Otera et al., 2010*). Instead, recent studies implicate FIS1 and its interacting protein TBC1D15 (*Onoue et al., 2013*) in mitochondrial degradation, specifically in PARKIN-dependent mitophagy (*Shen et al., 2014*; *Yamano et al., 2014*). Similar to *Fis1* deletion, *Tbc1d15* knockdown efficiently blocked mitophagy and decreased LC3 and p62 localization to mitochondria (*Figure 3C–E*). Expression of shRNA-resistant *Tbc1d15* in these cells restored red puncta formation (*Figure 4—figure supplement 1B,C*). Because depletion of either FIS1 or TBC1D15 blocked mitophagy and abolished P62 localization to mitochondria, we tested whether P62 is required for mitophagy. Cells knocked down for *p62*, as well as *p62* knockout cells, were deficient for OXPHOS-induced mitophagy and showed reduced mTurquoise2-LC3B localization to mitochondria (*Figure 3C,E*; *Figure 3—figure supplement 1A–B*). Expression of mTurquoise2-*p62* restored red puncta formation in *p62* knockout cells, and expression of shRNA-resistant *p62* restored red puncta formation in *p62* shRNA expressing cells, consistent with a role for P62 in

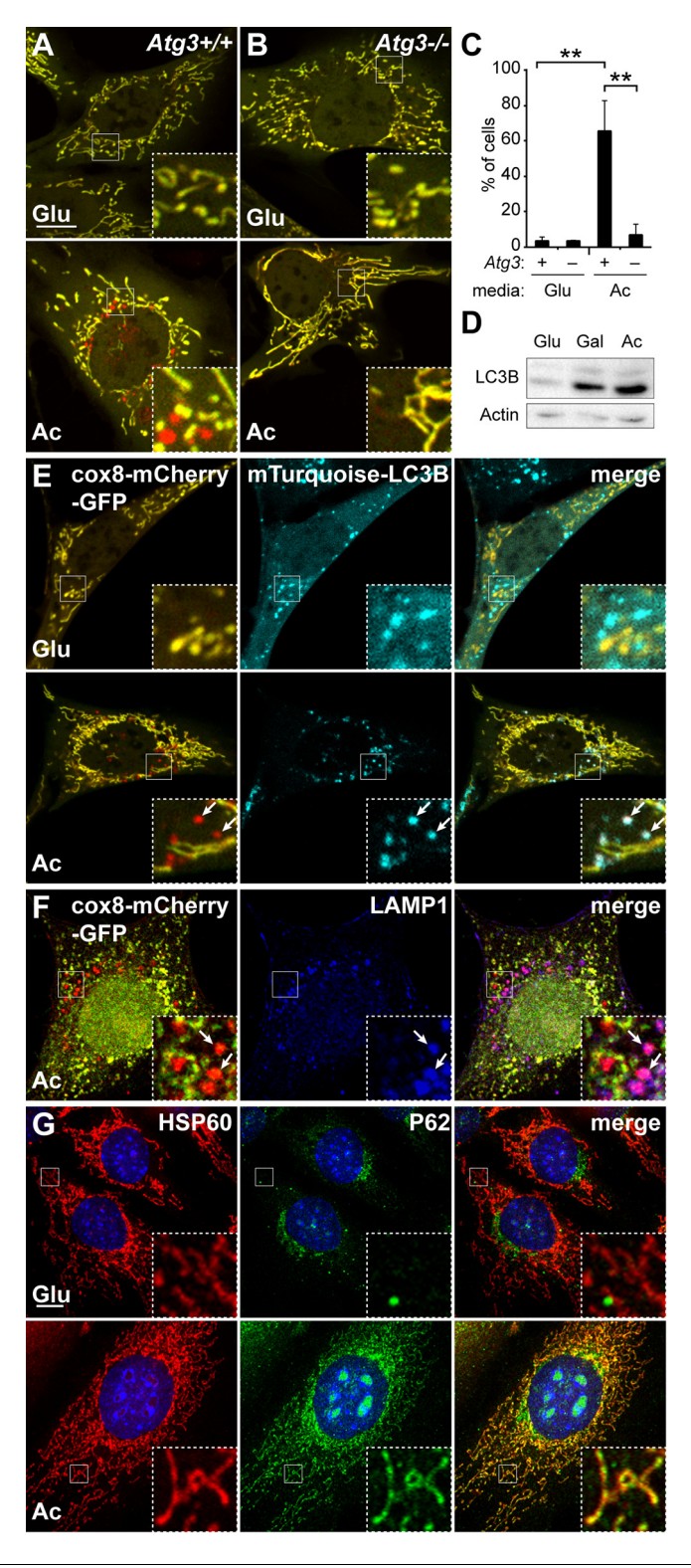

**Figure 2.** Induction of mitophagy by OXPHOS-inducing medium. Mitophagy was examined in cells stably expressing Cox8-EGFP-mCherry. Wild-type (**A**) or *Atg3* knockout mouse embryonic fibroblasts (MEFs) (**B**) were grown in Glucose (Glu) or Acetoacetate (Ac) containing medium for 4 days and then imaged by fluorescence microscopy. The red puncta in the bottom panel of (**A**) represent mitochondrial contents within acidic compartments. (**C**) Quantification of red-only puncta. Error bars indicate SD of three biological replicates,

*Figure 2 continued on next page*

*Figure 2 continued*

**p<0.01, p=0.0039 (Atg3+/+ Glu vs. Ac), p=0.0052 (Atg3+/+ vs. Atg3 -/-) (Student's *t*-test). (**D**) Western blot analysis of LC3B expression in MEFs cultured in the indicated medium. The lower band is lipidated LC3B. Actin is a loading control. (**E**) Co-localization of LC3B with red puncta. MEFs expressing cox8-EGFP-mCherry and mTurquoise2-LC3B were grown in the indicated medium and imaged by fluorescence microscopy. Arrows indicate examples of mTurquoise2-LC3B co-localization with red mitochondrial puncta. (**F**) Co-localization of LAMP1 with red puncta. MEFs stably expressing cox8-EGFP-mCherry were grown in acetoacetate-containing medium and immunostained with anti-Lamp1 antibody (blue). Arrows indicate red mitochondrial puncta that co-localized with LAMP1. Scale bar in (**A**) is 10 μm and applies to (**A–F**). (**G**) Co-localization of p62 with mitochondria. MEFs were grown in the indicated medium and immunostained with anti-p62 (green) and anti-HSP60 (red, mitochondrial marker). Error bars indicate SD. Scale bar, 10 μm.

OXPHOS-induced mitophagy (*Figure 3—figure supplement 1C*, *Figure 4—figure supplement 1B*). Taken together, these results place FIS1 and TBC1D15 upstream of P62 in promoting autophagic engulfment of mitochondria.

## PARKIN and MUL1 coordinately regulate OXPHOS-induced mitophagy

Because PINK1 and PARKIN are central components of the most widely studied pathway for mitophagy (*Pickrell and Youle, 2015*), we tested the role of these molecules in our mitophagy assay. *Pink1-/-* cells showed a substantial reduction in OXPHOS-induced mitophagy (*Figure 4A–B*). However, *Parkin* knockout MEFs had normal mitophagy (*Figure 4A–B*), a surprising observation given that PINK1 is known to operate upstream of PARKIN (*Clark et al., 2006*; *Park et al., 2006*; *Yang et al., 2006*). This observation suggests that another molecule may compensate for the loss of PARKIN. Recently, the mitochondrial E3 ligase MUL1 (MULAN/MAPL), has been shown to act parallel to the PINK1/PARKIN pathway in ubiquitination and proteasomal degradation of mitofusin (*Yun et al., 2014*). We hypothesized that MUL1 might work in parallel with PARKIN in OXPHOS-induced mitophagy, such that its presence would maintain mitophagy in the absence of PARKIN. Indeed, knockdown of *Mul1* by either of two independent shRNAs in the *Parkin* knockout cell abolished mitophagy (*Figure 4A–B*; *Figure 4—figure supplement 1A–C*). In contrast, knockdown of *Mul1* alone did not inhibit mitophagy. Inhibition of mitophagy due to loss of PINK1 or PARKIN/ MUL1 prevented co-localization of LC3 with mitochondria (*Figure 4C*). These results reveal that MUL1 and PARKIN have redundant functions in mitophagy. We found a similar redundancy of MUL1 and PARKIN function in mitophagy induced by depolarization of mitochondria with CCCP (*Figure 4—figure supplement 1D*)

Mitochondria from cells grown in OXPHOS media are ubiquitinated ~six-fold more than cells grown in glycolytic media (*Figure 4D,E*). Loss of MUL1 or PARKIN alone had modest or no effect on the induction of mitochondrial ubiquitination under OXPHOS conditions. However, loss of both MUL1 and PARKIN, or PINK1 alone, substantially reduced the ubiquitination of mitochondria (*Figure 4D,E*). Taken together, these data suggest that MUL1 and PARKIN act in concert to ubiquitinate mitochondrial substrates, and that a threshold level of ubiquitination may be required to trigger mitophagy under OXPHOS conditions. The level of mitochondrial ubiquitination is known to dynamically regulate mitophagy (*Bingol et al., 2014*; *Cornelissen et al., 2014*).

## Mitophagy genes are required for elimination of paternal mitochondria in embryos

With these molecular insights from the cellular assay, we re-visited the embryonic system to test whether the same pathway is involved in elimination of paternal mitochondria. We found that embryos expressing shRNA against *p62*, *Tbc1d15*, or *Pink1* showed strong suppression of paternal mitochondrial loss, compared to embryos expressing a non-targeting shRNA (*Figure 5A*). When these mitophagy genes were knocked down, the majority of embryos retained substantial paternal mitochondria at 84 hr post-fertilization (*Figure 5B*). In contrast, less than 20% of embryos containing non-targeting shRNA retained significant paternal mitochondria, with the majority of embryos showing complete loss of paternal mitochondria. Depletion of either *Parkin* or *Mul1* alone modestly reduced paternal mitochondrial elimination, but depletion of both had a severe and highly

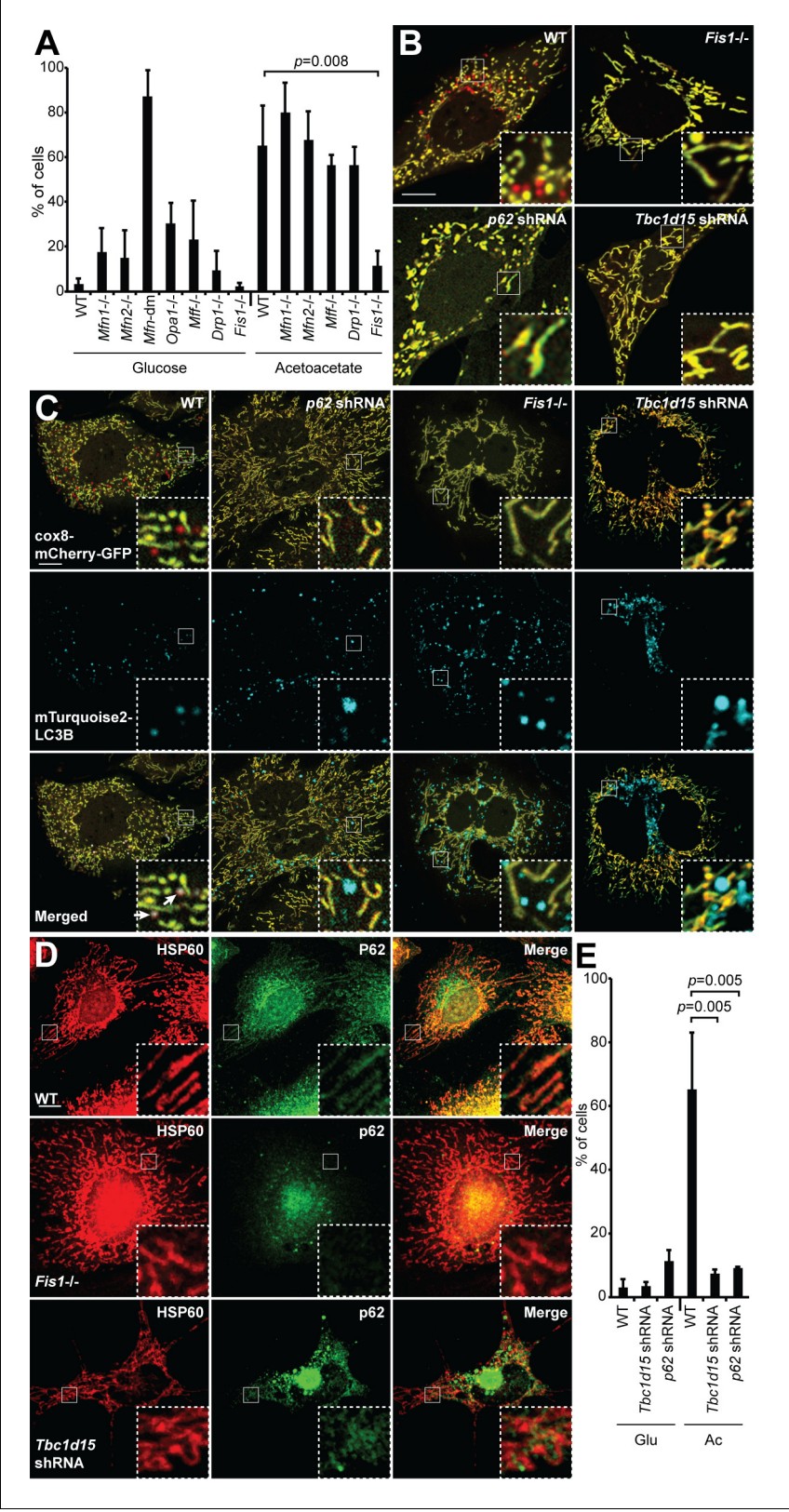

**Figure 3.** Mitophagy under OXPHOS-inducing conditions requires FIS1, TBC1D15, and p62. (**A**) Mitophagy in cells with mutations in mitochondrial dynamics genes. MEFs of the indicated genotype were cultured in glucose or acetoacetate medium, and mitophagy was quantified using the Cox8-EGFP-mCherry marker. Neither *Mfn*-dm cells

*Figure 3 continued on next page*

*Figure 3 continued*

nor *Opa1-/-* cells were viable in acetoacetate-containing medium. Error bars indicate SD of three biological replicates, p=0.0078 (Student's *t*-test. (B) MEFs stably expressing Cox8-EGFP-mCherry were grown in acetoacetate containing medium and then imaged by fluorescence microscopy. *p62* and *Tbc1d15* shRNAs were introduced by retroviral infection. (C) Co-localization of mTurquoise2-*LC3B* with mitochondria. MEFs were grown in acetoacetate containing medium. Note that mTurquoise2 puncta localize to mitochondrial puncta (arrows) only in WT cells. (D) Co-localization of P62 with mitochondria. MEFs were grown in acetoacetate containing medium and immunostained with anti-P62 (green) and anti-HSP60 (red). (E) Quantification of red-only puncta in WT cells and cells containing shRNA against *Tbc1d15* or *p62* cultured in glucose (Glu) or acetoacetate (Ac) medium. Error bars indicate SD of three biological replicates, p=0.0048 (*Tbc1d15*), p=0.0053 (*p62*) (Student's *t*-test).

The following figure supplement is available for figure 3:

**Figure supplement 1.** *p62* knockout cells have defective OXPHOS-induced mitophagy.

significant effect. Over 60% of *Parkin/Mul1*-depleted embryos showed retention of paternal mitochondria at 84 hr (*Figure 5A–B*, *Figure 5—source data 1*).

Although FIS1 is a key molecule in the OXPHOS-induced mitophagy pathway, the relevant FIS1 molecules are likely to be contributed by the sperm and not the egg. Our shRNA approach can only knockdown proteins synthesized within the embryo. To circumvent this issue, we developed a dominant negative version of FIS1 (FIS1-DN) that lacks the C-terminal transmembrane domain that is essential for mitochondrial outer membrane localization. Retroviral overexpression of the cytosolic FIS1-DN protein in MEFs strongly inhibits OXPHOS-induced mitophagy (*Figure 5—figure supplement 1A–C*). When FIS1-DN was expressed in embryos, we found that loss of paternal mitochondria was strongly inhibited, with the majority retaining substantial paternal mitochondria (*Figure 5C–D*, *Figure 5—source data 1*).

The signal for selective degradation of paternal mitochondria in mammals is unknown, but some other forms of mitophagy are triggered by loss of mitochondrial membrane potential. Using the cationic dye TMRE (tetramethylrhodamine ethyl ester), we found robust staining of sperm isolated from the caudal epididymis of *PhAM* male mice, indicating intact mitochondrial membrane potential (*Figure 6A*). At 18 hr after fertilization, paternal mitochondria remained in a linear cluster in the embryo and stained robustly with TMRE. However, over the next 36 hr, paternal mitochondria gradually lost TMRE staining, such that at 48 hr and later, nearly all paternal mitochondria failed to stain with TMRE (*Figure 6B–C*). In the same experiment, maternal mitochondria always maintained TMRE staining, indicating that there is selective loss of membrane potential in paternal mitochondria

We also examined whether paternal mitochondria fused with maternal mitochondria. To assess mitochondrial fusion, we utilized photo-conversion of Dendra2. We generated embryos in which either maternal mitochondria or paternal mitochondria were labeled with Dendra2. At 36 hr after fertilization, we photo-converted a subset of mitochondria and tracked their fate by confocal microscopy. At 60 hr post-fertilization, the photo-converted signal in maternally labeled embryos had spread widely into other mitochondria, resulting in a dramatic reduction in mean pixel intensity (*Figure 6—figure supplement 1A–B*). In contrast, the photo-converted signal from paternal mitochondria did not diffuse and clearly did not undergo fusion with other mitochondria in the embryo. This segregation of paternal mitochondria is likely to be important for their eventual degradation.

## Discussion

Our results provide two major insights about mitophagy in mammals. First, we find in two biological systems—OXPHOS-induced mitophagy in cultured cells and paternal mitochondrial elimination in pre-implantation embryos—that PARKIN and MUL1 work synergistically to promote degradation of mitochondria by autophagy. Previous work showed that PARKIN and MUL1 have partially redundant roles in controlling the ubiquitin-dependent degradation of mitofusin (*Yun et al., 2014*). Our results show that this collaboration extends to the process of mitophagy. In MEFs, we find that both PARKIN and MUL1 regulate the levels of ubiquitin on mitochondria in response to OXPHOS conditions, and removal of both is necessary to cause a substantial reduction of ubiquitination. In a mitophagy

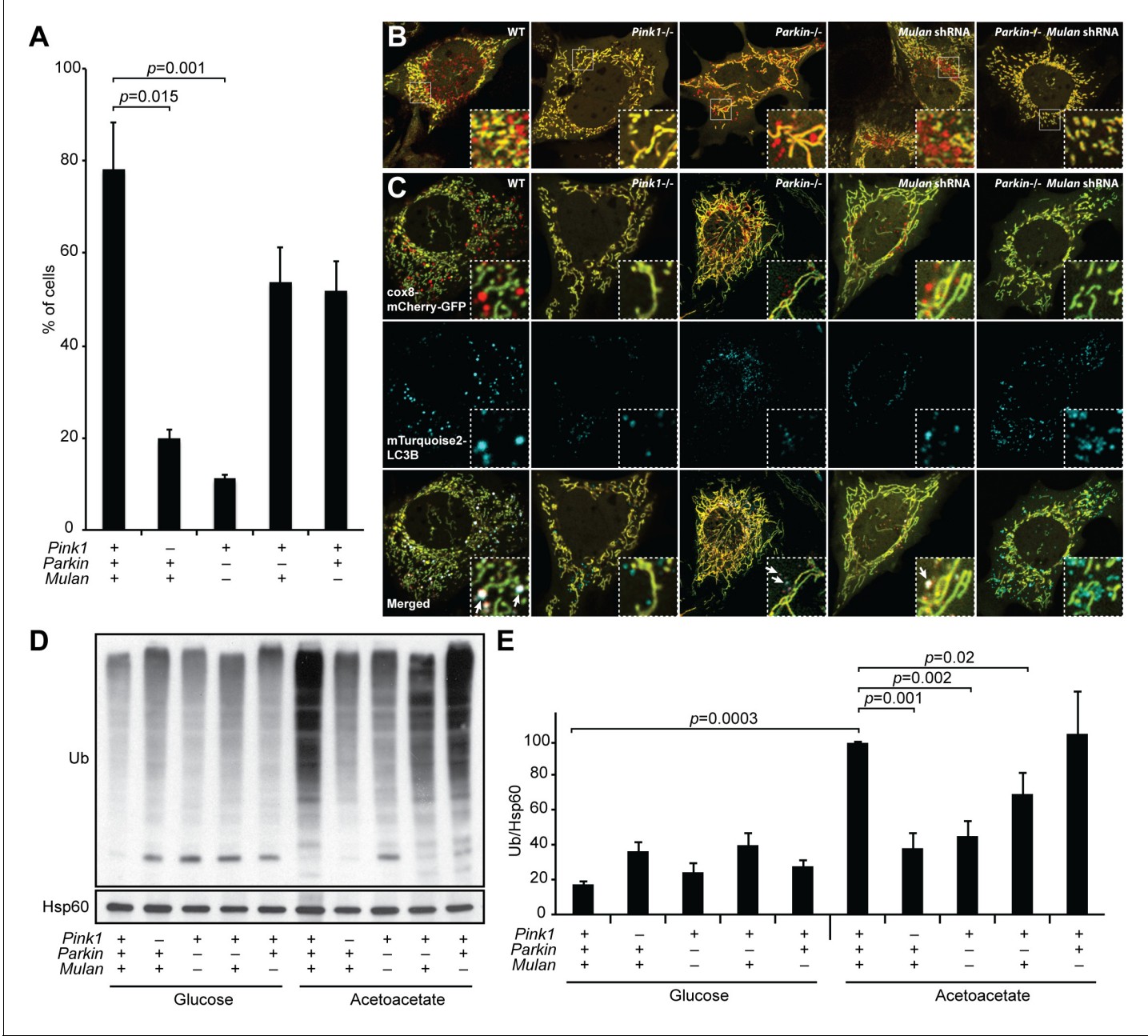

**Figure 4.** MUL1 and PARKIN have redundant functions in OXPHOS-induced mitophagy. (**A**) Quantification of red-only puncta in cells grown in acetoacetate-containing medium. Presence (+) or absence (-) of *Pink1*, *Parkin*, or *Mul1* is indicated. Error bars indicate SD of three biological replicates, p=0.015 (*Pink1*), p=0.0011 (*Parkin-/- Mulan* shRNA) (Student's *t*-test). (**B**) Mitophagy in wild-type and mutant cells. Cells stably expressing Cox8-EGFP-mCherry were grown in acetoacetate-containing medium and imaged by fluorescence microscopy. (**C**) Co-localization of LC3B with mitophagy intermediates. Wild-type and mutant cells were retrovirally transduced with mTurquoise2-LC3B, grown in acetoacetate-containing medium and imaged by fluorescence microscopy. Examples of LC3B co-localization with mitophagy intermediates are indicated by arrows. (**D**) Accumulation of polyubiquitinated proteins in mitochondria. Cells were grown in the indicated medium, and mitochondria were isolated by differential centrifugation. Mitochondrial lysates were analyzed by Western blot for pan-Ubiquitin. HSP60 is a loading control. (**E**) Quantification of polyubiquitinated proteins in mitochondria. Three independent experiments were quantified by densitometry and averages are shown. Ubiquitin level was normalized to HSP60. Error bars indicate SD, p=0.0003 (WT Glu vs. Ac), p=0.0011 (*Pink1-/-*), p=0.0016 (*Parkin-/- Mulan* shRNA), p=0.0206 (*Parkin-/-*) (Student's *t*-test).

The following figure supplement is available for figure 4:

**Figure supplement 1.** Defective mitophagy in *Parkin/Mul1*-deficient cells.

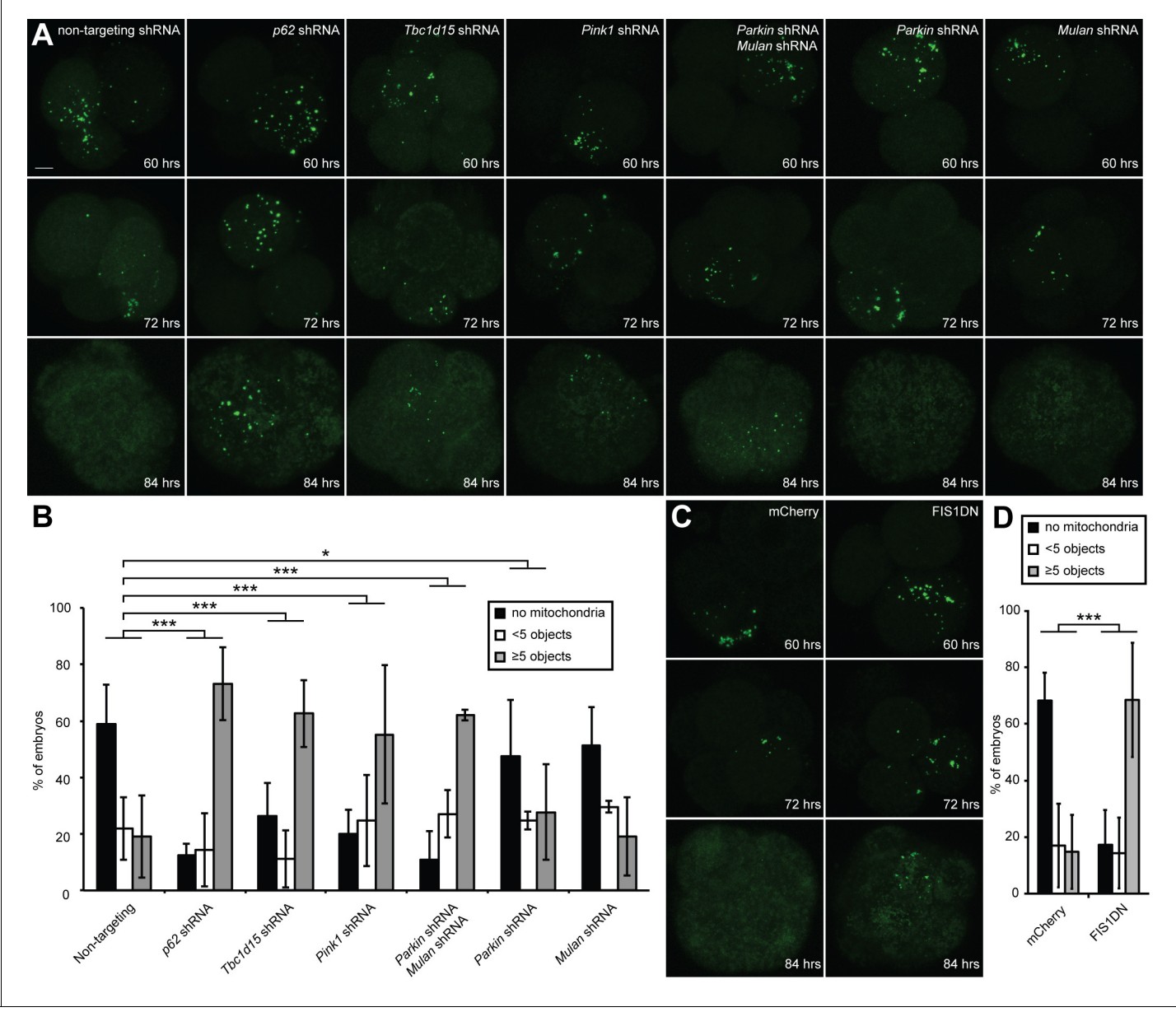

**Figure 5.** Clearance of paternal mitochondria in preimplantation embryos requires mitophagy genes. (A) Impaired elimination of paternal mitochondria upon inhibition of mitophagy genes. Embryos were injected with lentivirus expressing shRNA against the indicated genes. The mitochondrial Dendra2 signal is shown for live embryos at 60, 72, and 84 hr after fertilization. Images are maximum intensity projections. Scale bar, 10 μm. (B) Quantification of paternal mitochondrial elimination at 84 hr post-fertilization. Maximum intensity z-projection images were analyzed encompassing the full embryo with z-slices overlapping. Embryos were scored as having no paternal mitochondria (black bar), less than five mitochondrial objects (white bar), or five or more mitochondrial objects (grey bar). Averages of at least three independent injection experiments are shown with 32–200 embryos quantified. Error bars indicate SD, *p<0.05; **p<0.01; ***p<0.001 (Chi-squared test). p-Values compare experimental embryos to control embryos with non-targeting shRNA. Chi-squared values: 75.386 (*Tbc1d15*), 155.784 (*p62*), 58.064 (*Parkin* shRNA, *Mulan* shRNA), 1.484 (*Mulan* shRNA), 8.074 (*Parkin* shRNA). (C) Clearance of paternal mitochondria in embryos expressing mCherry (control) or *Fis1*-DN. Same scale as (B). (D) Quantification of 84 hr results from (D). Error bars indicate SD. ***p<0.001 (Chi-square test). p-Values compare experimental embryos to mCherry control embryos. Chi-squared value: 125.584.

The following source data and figure supplement are available for figure 5:

**Source data 1.** Source data for *Figure 5B and D*.

**Figure supplement 1.** Inhibition of OXPHOS-induced mitophagy by dominant negative FIS1.

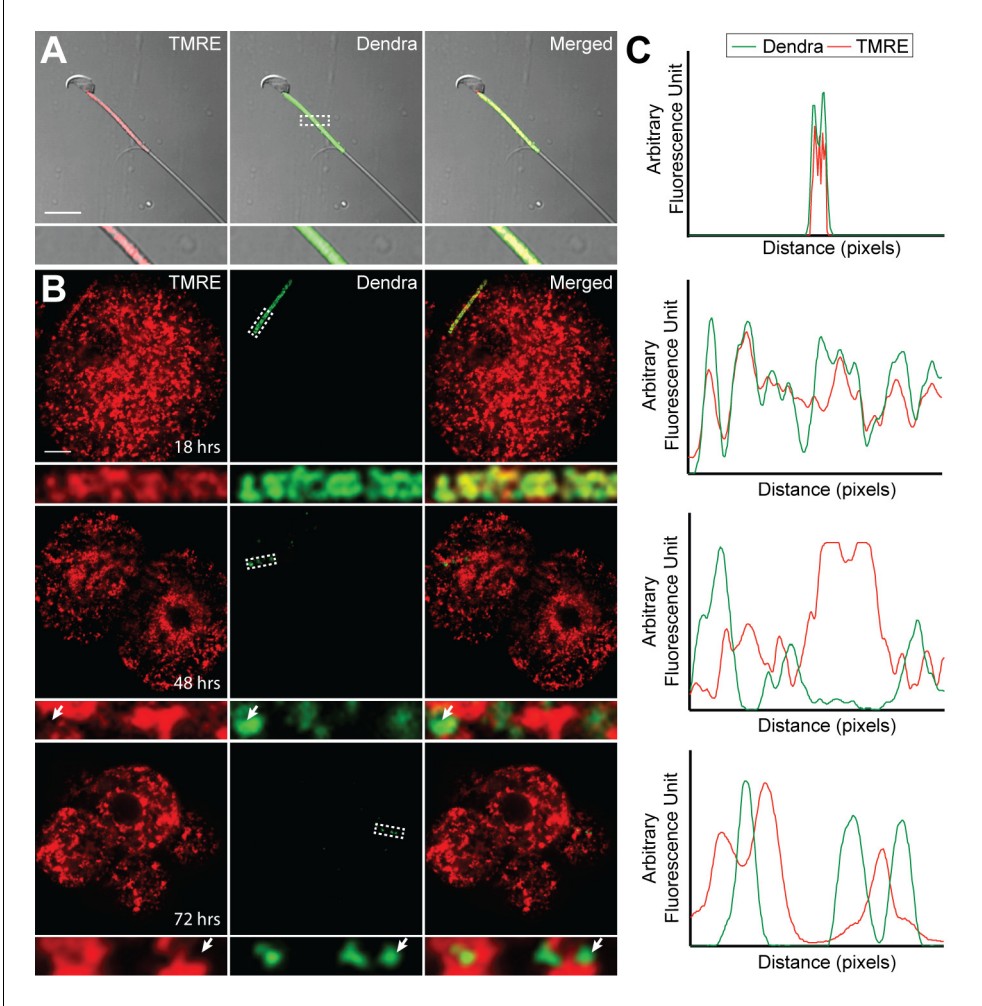

**Figure 6.** Loss of membrane potential in paternal mitochondria after fertilization. (**A**) Mitochondrial membrane potential in live sperm cell. Spermatozoa were isolated from the cauda epididymis of a *PhAM* mouse, stained with 20 nM TMRE, washed, and imaged by fluorescent microscopy. Red signal is TMRE; green signal is mito-Dendra2. The boxed region is enlarged below. Scale bar, 10 μm. (**B**) Membrane potential of paternal mitochondria in early embryos. Embryos, generated by mating wildtype females with *PhAM* males, were collected at 12 hr after fertilization and cultured in vitro. At 18, 48, or 72 hr after fertilization, the embryos were incubated in 20 nM TMRE, washed, and imaged by fluorescent microscopy. Dashed box indicates region enlarged below. Arrows indicate examples of mito-Dendra2-positive spots lacking TMRE signal. Scale bar, 10 μm. (**C**) Fluorescence line analysis of the boxed regions in (**A**) and (**B**). Each plot measures the TMRE and mito-Dendra2 signals along a one-pixel width line through the center of the boxed region. Note that the mito-Dendra2 and TMRE signals are co-incident at 18 hr after fertilization but not at 48 or 72 hr.

The following figure supplement is available for figure 6:

**Figure supplement 1.** Fusion activity of maternal mitochondria versus paternal mitochondria in the early embryo.

assay where PARKIN is overexpressed, polyubiquitination of mitochondrial outer membrane proteins leads to their proteasomal degradation, which in turn is required for turnover of mitochondria by autophagy (*Chan et al., 2011*).

The redundant function of MUL1 likely explains why PARKIN knockout mice show surprisingly mild and inconsistent mitochondrial phenotypes (*Palacino et al., 2004*; *Perez and Palmiter, 2005*). Similarly, the redundant role of MUL1 may also explain why, in *Drosophila*, PARKIN is dispensable

for paternal mitochondrial elimination (*Politi et al., 2014*). In future work, this insight may help to uncover the in vivo functions of PARKIN.

Second, we show that mitophagy is likely to be the mechanism underlying the elimination of paternal mitochondria in the early mouse embryo. It is unclear whether MEFs cultured under OXPHOS conditions bear any physiological relation to the early embryo. Nevertheless, we find that the genetic requirements for removal of paternal mitochondria in the embryo mirror those of MEFs undergoing mitophagy in response to OXPHOS induction. Although previous studies had shown that paternal mitochondria in mouse embryos co-localized with autophagy markers (*Al Rawi et al., 2011*), the functional relevance of these localization studies has been challenged and a passive mechanism for loss of paternal mitochondria has been proposed (*Carelli, 2015*; *Luo et al., 2013*). By identifying several molecules necessary for paternal mitochondrial elimination, our studies provide functional evidence for the role of mitophagy in this process.

Because we find that paternal mitochondria lose membrane potential shortly after entering the oocyte, it is tempting to speculate that this membrane depolarization may be the trigger for mitochondrial degradation. Previous studies indicate that PARKIN is recruited to mitochondria upon membrane depolarization (*Narendra et al., 2008*, *2010*), and our results also suggest that PARKIN and MUL1 work together to degrade mitochondria that are depolarized (*Figure 4—figure supplement 1D*). However, we do not have direct evidence that membrane depolarization has a functional role in paternal mitochondrial degradation.

Although uniparental inheritance of mitochondria is nearly universal in animals, its physiological function is mysterious and difficult to address. One recent idea is that uniparental inheritance of mitochondria ensures that offspring contain only one haplotype of mtDNA. When mice with approximately equal proportions of two wild-type haplotypes of mtDNA were generated, they were found to have behavioral and cognitive abnormalities compared to homoplasmic counterparts (*Sharpley et al., 2012*). However, it is unclear to what extent this experimental result is relevant for a case in which paternal mitochondria were not eliminated. Sperm contain many fewer mitochondria (at least a thousand fold) compared to the oocyte, and therefore, the ensuing heteroplasmy levels would be very low. The identification of molecules essential for paternal mitochondrial elimination may facilitate further examination of this issue.

## Materials and methods

### Antibodies

The following commercially available antibodies were used: anti-Actin (Mab1501R, Millipore), anti-HSP60 (SC-1054, Santa Cruz Biotech), anti-LAMP1 (1D4B, Developmental Studies Hybridoma Bank), anti-P62 (PM045, MBL), anti-LC3B (2775 s, Cell Signaling), anti-c-Myc (C3956, Sigma), anti-Ubiquitin (P4D1, Cell Signaling), anti-PINK1 (75488, Abcam), anti-TBC1D15 (121396, Abcam), anti-PARKIN (15954, Abcam), and anti-MUL1 (HPA017681, Sigma).

For Western analysis, densitometry was done using ImageJ. The intensity of the ubiquitin signal was normalized to that of HSP60, and the average of three separate experiments was taken.

### Immunostaining

For immunofluorescence experiments, cells were fixed with 10% formalin, permeabilized with 0.1% Triton X-100 and stained with the primary antibodies listed above and with the following secondary antibodies: goat anti-mouse Alexa Fluor 633, donkey anti-goat Alexa Fluor 546, goat anti-rabbit Alexa Fluor 488, goat anti-rabbit Alexa Fluor 633 (Invitrogen, Carlsbad, CA). When used, DAPI (d1306, Invitrogen) was included in the last wash.

### shRNA virus design and production

For experiments in MEFs, the retroviral vector pRetroX-H1, which contains the H1 promoter, was used to express shRNAs. shRNAs were cloned into the BglII/EcoRI sites. For embryo injection experiments, a third-generation lentiviral backbone was used to express shRNAs. The lentiviral vector FUGW-H1 (*Fasano et al., 2007*) was modified by replacing the GFP reporter gene with mCherry and changing the shRNA cloning sites from Xba/SmaI to BamHI/EcoRI, generating FUChW-H1. For

dual knockdown experiments in embryos, a second H1 promoter was added, along with XbaI/NheI cloning sites 3' to the original H1 promoter, generating FUChW-H1H1.

The shRNA target sequences were:
*p62*: TGGCCACTCTTTAGTGTTTGTGT
*Tbc1d15*: GTGAGCGGGAAGATTATAT
*Mul1* sh1: GAGCTAAGAAGATTCATCT
*Mul1* sh2: GAGCTGTGCGGTCTGTTAA
*Pink1*: GGCTGACAGGCTGAGAGAGAA
*Parkin*: CCTCCAAGGAAACCATCAA
Non-targeting: GACTAGAAGGCACAGAGGG

Lentiviral vectors were cotransfected into 293T cells with plasmids pMDLG/pRRE, pIVS-VSVG, and pRSV-Rev. Retroviral vectors were cotransfected into 293T cells with plasmids pVSVG and pUMVC. All transfections were done using calcium phosphate precipitation. For microinjection, virus was collected, filtered, concentrated by ultracentrifugation at 25,000 rpm for 2 hr, resuspended in PBS and stored at −80°C as described previously (*Lois et al., 2002*; *Pease and Lois, 2006*). Viral titers were measured by infecting MEFs with serial dilutions of viral preparations, followed by flow cytometric analysis after 48 hr. Virus was used at $1 \times 10^7$ transducing units/μL.

## Embryo microinjection

All mouse work was done according to protocols approved by the Institutional Animal Care and Use Committee at the California Institute of Technology. For each experiment, four C57/Bl6J wild-type female mice at 21–25 days old were superovulated by hormone priming as described previously (*Pease and Lois, 2006*), and then each was caged with a *PhAM* male (*Pham et al., 2012*) (RRID: IMSR_JAX:018397). After euthanization of females by $CO_2$ asphyxiation, the embryos were harvested and placed in M2 medium (MR-015-D, Millipore) at 12 hr after fertilization as described in (*Pease and Lois, 2006*). Approximately 60 to 100 embryos were collected per experiment. Embryos were divided into two equal groups and microinjected with 10 to 100 pl of viral stock into the perivitelline space as described in (*Lois et al., 2002*; *Pease and Lois, 2006*). Embryos were washed with KSOM+AA medium (MR-106-D, Millipore) and cultured in that medium covered by oil (M8410, Sigma) at 37°C and 5% $CO_2$. For each construct, at least three separate microinjection sessions were performed. In preparation for imaging, embryos were transferred to 10 μl droplets of KSOM+AA medium on glass-bottom dishes (FD35-100, World Precision Instruments).

## Imaging and quantification

All images were acquired with a Zeiss LSM 710 confocal microscope with a Plan-Apochromat 63X/ 1.4 oil objective. All live imaging was performed in an incubated microscope stage at 37°C and 5% $CO_2$. The 488 nm and 561 nm laser lines were used to excite cox8-EGFP-mCherry and imaging was done in line mode to minimize movement of mitochondria between acquisition of each channel. The 405 nm laser line was used to excite mTurquoise2 and DAPI. Alexa 488, Alexa 546, and Alexa 633, conjugated dyes were excited by the 488 nm laser, 561 nm laser, and the 633 nm laser, respectively.

In experiments tracking paternal mitochondrial degradation, all viable embryos from each experiment were imaged. Only embryos that were fragmented, lysed, or developmentally delayed were not imaged. The top and bottom of the embryo was set as the top and bottom z slices for z-stack image acquisition. Optical slices were acquired at 1.1 μm thickness, and z stacks were oversampled at 0.467 μm to ensure that all mitochondria were captured. Maximum intensity projections were created with Zen 2009 software and used for quantification.

For quantification of paternal mitochondria, control and experimental embryo images were randomized and counted blind. The number of mitochondria within each embryo was counted manually. In cases where two or more mitochondria were clustered together and could not be definitely resolved as distinct objects with separable borders, the cluster was counted as one object. Each maximum intensity z-projection was categorized as having either no mitochondria, less than five mitochondrial objects, or five or more mitochondrial objects. Embryos from four females were pooled per experiment, and three or more independent replicate experiments were averaged.

For photo-conversion of Dendra2, a region of interest was illuminated with the 405 nm line (4% laser power) for 90 bleaching iterations. The 488 nm laser line (5% laser power) and the 561 nm laser

line (6.5% laser power) were used to excited Dendra2 in the unconverted state and photo-converted state, respectively. The pinhole used was 159 microns. Bandpass filters were used for detection of unconverted and photo-converted Dendra2 from 494 to 547 nm and 566 to 735 nm, respectively. The mercury lamp was not used to avoid inadvertent photoconversion.

For quantification of photo-converted Dendra2, maximum intensity z-stacks encompassing the entire embryos were analyzed in Matlab. For total intensity measurement, positive pixels were defined as those having an intensity greater than 10 (a low threshold designed to remove background), and the sum of these pixel intensities was calculated. For mean intensity measurement, this sum was divided by the total number of positive pixels.

Images were cropped when appropriate, and image contrast and brightness were globally adjusted in Photoshop (Adobe). Replicates are as indicated in figure legends.

## Isolation of spermatocytes

Sperm were isolated from 4-month-old *PhAM* male mice. Longitudinal cuts were made in the cauda epididymis, and the tissue was incubated in PBS at 37°C to enable motile, mature sperm to swim out.

## Membrane potential measurements

TMRE fluorescence was used to monitor mitochondrial membrane potential in spermatocytes and embryos. Samples were loaded with 20 nM TMRE for 20 min at 37°C and then washed into PBS (spermatocytes) or KSOM+AA (embryos). Samples were imaged live. Line analysis was performed using ImageJ.

## Isolation of mitochondria

Mitochondria were isolated by differential centrifugation. Cells were washed in PBS, collected by scraping in isolation buffer (220 mM mannitol, 70 mM sucrose, 80 mM KCl, 5 mM MgCl$_2$, 1 mM EGTA, 10 mM K$^+$HEPES, pH7.4, and HALT protease inhibitors), and lysed on ice. Lysates were cleared of cell debris and nuclei with four 600 *g* spins. A crude mitochondrial fraction was isolated with a 10,000 *g* spin for 10 min and washed three times in isolation buffer.

## Retroviral expression constructs

The Cox8-EGFP-mCherry retroviral vector (kindly provided by Drs. Prashant Mishra and Anh Pham) consists of the *Cox8* mitochondrial targeting sequence placed N-terminal to an EGFP-mCherry fusion. To clone mTurquoise2 fusion proteins, mTurquoise2 was amplified from pmTurquoise2-Mito (Addgene plasmid # 36208, Dorus Gadella, [*Goedhart et al., 2012*]). Human LC3B was amplified from pFCbW-EGFP-LC3. Mouse *p62* was amplified from pMXs-puro GFP-*p62* (Addgene plasmid # 38277, Noboru Mizushima, [*Itakura and Mizushima, 2011*]). mTurquoise2 fusion proteins were cloned into the retroviral vector, pBABEpuro. The FIS1 dominant negative construct was cloned into pBABEpuro and consists of amino acids 1–121 of mouse FIS1, with 9 Myc tags at the N-terminus. The corresponding control construct consists of mCherry cloned into the pBABEpuro vector. All plasmids were verified by DNA sequence analysis. Stable cell lines were generated by retroviral infection followed by selection with 2 µg/µl puromycin.

## Cell culture

MEFs were maintained in Dulbecco's Modified Eagle's Medium (DMEM) supplemented with 10% fetal bovine serum (FBS), 100 U/mL penicillin and 100 U/mL streptomycin at 37°C and 5% CO2. Glucose and acetoacetate containing media were made as previously described (*Mishra et al., 2014*). For mitophagy experiments, cells were plated on Nunc Lab-Tek II Chambered Coverglass slides (155409, Thermo) in DMEM-based media. After cells had adhered, they were washed with PBS and glucose- or acetoacetate-containing medium was applied, after which cells were allowed to grow for 4 days and then imaged. Because cells grow more slowly in acetoacetate medium, a four-fold excess of cells was plated relative to glucose medium so that both samples were at the same density on the day of imaging.

## Cell lines

The cells used included: *Atg3*-null MEFs (*Sou et al., 2008*) (kindly provided by Yu-Shin Sou and Masaaki Komatsu), *p62*-null MEFs (*Ichimura et al., 2008*) (kindly provided by Shun Kageyama and Masaaki Komatsu), *Pink1*-null, *Parkin*-null (both kindly provided by Clement Gautier and Jie Shen), and *Drp1*-null (*Ishihara et al., 2009*) (kindly provided by Katsuyoshi Mihara). *Mfn1*-null (ATCC Cat# CRL-2992, RRID:CVCL_L691), *Mfn2*-null (ATCC Cat# CRL-2993, RRID:CVCL_L693), *Mfn*-dm (ATCC Cat# CRL-2994, RRID:CVCL_L692), *Opa1*-null (ATCC Cat# CRL-2995, RRID:CVCL_L694), *Mff*-null, *Fis1*-null MEFs have been described previously (*Chen et al., 2005*; *Losón et al., 2013*). The identity of MEF cell lines was authenticated by PCR genotyping of the relevant gene. Cell lines were negative for mycoplasma by DAPI staining.

## Acknowledgements

We are grateful to Shirley Pease (Director, Transgenic Core at Caltech) for training and advice on embryo injection. We thank Katherine Kim for preliminary work with *MUL1* knockdown experiments, Kurt Reichermeier for advice on the ubiquitin assay, Ruohan Wang for technical assistance with *p62* overexpression, and Hsiuchen Chen for advice on animal work. RR is supported by an NIH NIGMS training grant (GM08042) and the UCLA Medical Scientist Training Program.

## Additional information

### Funding

| Funder | Grant reference number | Author |
| --- | --- | --- |
| National Institute of General Medical Sciences | GM08042 | Rebecca Rojansky |
| National Institutes of Health | GM119388 | David C Chan |
| National Institutes of Health | GM083121 | David C Chan |

The funders had no role in study design, data collection and interpretation, or the decision to submit the work for publication.

### Author contributions

RR, Conception and design, Acquisition of data, Analysis and interpretation of data, Drafting or revising the article; M-YC, Acquisition of data, Analysis and interpretation of data; DCC, Conception and design, Analysis and interpretation of data, Drafting or revising the article

### Author ORCIDs

David C Chan, http://orcid.org/0000-0002-0191-2154

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
