## [Decision Letter]

Thank you for submitting your article "Elimination of paternal mitochondria in mouse embryos occurs through autophagic degradation dependent on PARKIN and MUL1" for consideration by *eLife*. Your article has been favorably evaluated by Sean Morrison as the Senior Editor and three reviewers, one of whom is a member of our Board of Reviewing Editors. The reviewers have opted to remain anonymous.

The reviewers have discussed the reviews with one another and the Reviewing Editor has drafted this decision to help you prepare a revised submission. The reviewers had some serious concerns but believe your work is conceptually important and would be willing to consider a revised manuscript if you are able to address their concerns. The reviewer comments are shown below in full to provide context, but the essential points you would have to address are described here.

One main difficulty that comes through the comments of all three reviewers is the challenge to assess the specificity of the mitophagy process you describe for paternal mitochondria. Are maternal mitochondria in zygotes also subjected to a similar process or is it specific to paternal mitochondria? This question may be addressed by using the same mutant mouse line but this time using female mutants crossed with wild-type males and while the number of maternal mitochondria may be large, you may be able to limit your quantitation by photoconverting Dendra in discrete areas of the zygote or by quantifying a fixed number of fields.

The use of glucose-free, acetoacetate medium also raised a number of questions. Not much is known about the effects of this medium on the biology of the cell beyond its action on OXPHOS. Could this medium have other effects that confound the interpretation of the results in MEFs? Are these culture conditions relevant to the more physiological conditions in the zygote? If the zygote has a more glycolytic metabolism, and the machinery identified here required an enhanced OXPHOS metabolism, how accurate can the identified machinery be? Clearly, some core elements such as Parkin and Mulan have been identified and are sufficient to modulate mitophagy in zygotes but these questions regarding the appropriateness and relevance of the culture medium must be addressed.

This led us to raise the question of membrane potential. Was the machinery of mitophagy that you identified validated under normal membrane potential? In the last paragraph of the Results section you indicate that zygotes show a time-dependent loss of membrane potential, suggesting that this trigger is important to the autophagy of paternal mitochondria. From your data you do not demonstrate that the loss of membrane potential is instrumental here. It is possible that the loss of membrane potential is an epiphenomenon, unrelated to the autophagy of paternal mitochondria, since the machinery of mitophagy you report herein has not been identified in the context of loss of membrane potential. It would thus be helpful to at least validate your identified mitophagy machinery under an induced membrane potential defect. Formal assessment in the zygotes of the membrane potential in maternal mitochondria should also be provided.

Your cellular and subcellular quantification methods were also questioned, and it would be important to better define how many elements were counted, how they were selected, and in how many cells from the embryos. Along this line it may be helpful to more clearly stress in the results that you counted all of the cells in each embryo and all paternal mitochondria in each cell. If, in some instances, this was not the case, then you must clarify how cells and/or mitochondria were selected and whether corrections were made for different number of cells per embryos of different ages. In addition, it was not clear what the level of detection is of Dendra-containing mitochondria in the microscope. Is the precision of your method and probe sufficient to confirm that one counted object equals one organelle? Also not clear from the manuscript was how you deal with detection of two mitochondria that are the fragmented/fissioned daughters of a parental organelle. Lastly, is it possible to photoconvert a single mitochondrion and follow its fate (sounds technically daunting but would be truly informative)?

As you consider revising your manuscript, please also provide more detailed information about the statistical treatment of the data as requested by reviewer #1. Also, it would also be important to provide further detail and additional controls for the knockdown experiments to assure that the correct targets were indeed silenced and the effects do not reflect off-target effects, as discussed by reviewer #2.

Finally, we would like to provide our editorial position with respect to two more points. First, with respect to the novelty of the study, as questioned by reviewer #3 ("The MEF experiments are convincing, but do not illustrate novel mechanisms"), it is our position that the novelty of the study does not reside in the "novel mechanism" of mitophagy, but in the demonstration that mitophagy (even if it uses a known mechanism) is involved in the elimination of paternal mitochondria.

Second, with respect to some of the mechanistic gaps, primarily pertaining to how and when paternal mitochondrial are recognized (e.g. reviewer #3, points #1, 2, and 6), these are valid questions but are beyond the scope of this specific study. However, a valuable approach to at least provide support to the specificity of the mechanism for the paternal mitochondria would be to compare intra- vs. inter-species mitophagy. Indeed, the phenomenon of paternal mitochondrial incompatibility applies only to intra-specific matings in that in inter-specific matings the paternal mitochondria survive, at least in mice (Kaneda et al., PNAS 92:4542, 2015). Consequently, an inter-specific mating should essentially abrogate all the relevant phenomena found here in the intra-specific mating (e.g. the decay of Dendra signal), thereby strongly supporting your conclusions.

Reviewer #1:

In this new study, the authors report on the fundamental, unresolved question of how the paternal mitochondria, which come from the spermatozoid and which enter the oocyte upon fertilization, are eliminated, leaving the blastocyst cells populated with only maternal mitochondria. In mammals, the debate remains as to whether paternal mitochondria are eliminated by stochastic dilution (related to the random partition of mitochondria during cell division) or mitophagy, and while these two processes are not mutually exclusive, it has been quite difficult to demonstrate a role for mitophagy in the destruction of paternal mitochondria. To clarify this fundamental issue, the authors used a mutant mouse in which all mitochondria are labelled with a fluorescent protein. By using a mutant male mouse crossed with a female wild-type mouse, the authors were able to readily identify and monitor paternal mitochondria in mouse zygotes. The authors confirmed that the number of paternal mitochondria declined over time and that by ~84 hrs post-fertilization, zygotes were devoid of any paternal mitochondria. As anticipated, using cells deficient in core autophagy factors such as ATG3 did stop embryo development and thus was not an unusable strategy to determine whether mitophagy was involved. Thus, the authors thought to use MEF cells to acquire further molecular insights into mitophagy to ultimately be able to target mitophagy rather than global autophagy, which they believed would be better tolerated by the embryo. Accordingly, they took a variety of MEF lines which were either constitutively deficient or were subjected to silencing strategy to knockout or knockdown key factors of mitochondrial fission/fusion as well as mitophagy such as Parkin and PINK1. To induce mitophagy, the authors exposed the different MEF lines to a glucose-free acetoacetate containing culture medium, and under these experimental conditions they found that: fusion/fission factors may modulate mitophagy but were not key factors except for *Fis1* which was found to be a regulator. They also found Parkin in a redundant manner with another E3 ligase, MUL1, which was also instrumental in mitophagy induced by this particular culture medium. Other known factors such as PINK1, P62, and *Tbc1d15* were all also implicated. With this information in hand, the authors went back to the zygotes and by emulating each of the molecular situations that were attenuating mitophagy in MEF, they consistently found that the elimination of paternal mitochondria was reduced. The authors concluded that their study provides compelling evidence that in mammalian cells, the elimination of paternal mitochondria in zygote and maternal mitochondria in MEFs used the same machinery of mitophagy and involved the cooperation between Parkin and MUL1.

This is quite an elegant study that provides an important demonstration about the involvement of mitophagy in the elimination of paternal mitochondria. The data are, for the most part, convincing, as are the interpretation and discussion of the results. Yet, a few weaknesses have been identified that reduced the enthusiasm for this work.

1) The authors demonstrate that targeting key factors of mitophagy does alter the decline of paternal mitochondria in zygote. However, since it is safe to assume that over ~84 hr primordial cells continue to divide, and that none of the proposed strategies abrogates but rather attenuates the decline, one cannot exclude a contribution of stochastic dilution in this process. One is thus left with the question of, if both processes are involved, what is the relative contribution of each to the elimination of paternal mitochondria? Unless overlooked, could the authors assess the decline by blocking cell division so that only mitophagy could be monitored?

2) While the strategy to find factors of mitophagy is original, it is somewhat confusing as it is not clear what this glucose-free acetoacetate containing medium does to the cell. While the authors claim that this culture medium promotes OXPHOS, the link between this effect and mitophagy is unclear. If as mentioned by the authors, loss of membrane potential is known to induce mitophagy, why did they not use this strategy to make their data perfectly comparable to those in most of the literature on the topic? Was there a problem on their hand in profoundly altering membrane potential?

3) Along this line, it is surprising that while the authors elected to move away from the membrane potential, in the last paragraph they examine this question in the embryos. This small piece of data reads as a last minute add-on with no connection to the rest of the study. Furthermore, these data do not demonstrate causality as they are strictly correlative and as such misleading.

4) It is also confusing to see that while the authors had to force a more OXPHOS-based metabolism to detect a meaningful factor of mitophagy, zygotes are cells that rather rely on glycolysis. If this is correct, there seems to be a disconnect in the logic of the experimental design.

5) More of the data are presented in a manner that the quantitative dimension of the observation is often difficult to assess. This manuscript is written more as a semi-quantitative/qualitative study, making it quite challenging to properly judge the significance of the results: are the data representing an exception or a rule?

6) Along this line, statistics are often too cursory presented. The authors must indicate for each of the tests used the N or DF, the results of the test and the actual p values. Also unclear is the number of technical and biological replicates that were used for each experiment.

7) There is an overall lack of technical details which prevents the reader from fairly evaluating the data.

Reviewer #1 (Additional data files and statistical comments):

As indicated above, there is an overall lack of details about the statistical treatment of the data.

Reviewer #2:

Rojansky and Chan confirm that genes already implicated in mitophagy, including PINK, p62, *Fis1* and TBC1D15, behave similarly in their ox/phos driven mitophagy assay. They also suggest that Parkin and Mulan (redundantly), PINK1 and p62 are required to eliminate paternal mitochondrial DNA in pre-implanted embyros up to 84 hours. While the pre-implantation and mitophagy assay are important tools to determine the factors necessary to eliminate paternal mitochondrial DNA, the conclusions could be better substantiated by taking the pre-implantation assay out longer if the point is to link the results to maternal inheritance. How PINK1 functions upstream of *Mul1* warrants further exploration. In addition, rescue experiments are required for both cell culture and embryo assays. Major methodological problems need to be addressed to build confidence in the conclusions.

1) In Figure 1 and Figure 5 the authors knock down various genes in the embryo using a single shRNA, but fail to show the efficiency of the knockdown. Assessment of knockdown is essential to make any conclusions regarding what genes are actually necessary for the elimination of paternal mitochondrial DNA.

2) Again, a key problem in Figure 3 and Figure 4 is that it is not shown to what extent the shRNAs knock down the expression of p62, *Mul1* or TBC1D15. Rescue experiments with Parkin and *Mul1*, for example, would shore up the conclusions and address off target *Mul1* shRNA effects.

3) It is not clear how accurate/quantitative the bar graphs of cell counting are (what is the lower cut off point for red dot number to count as mitophagy?) because in the one western blot shown (Figure 4) the ubiquitination in the *Mul1* shRNA lane 10 is not less than wild type in lane 6 whereas Parkin loss shows a clear effect – inconsistent with the two right bars in the cell counting results in Figure 4 main conclusion of the manuscript. A more quantitative approach that would assess total red dot number, such as FACS, would likely improve confidence in the comparisons. Or other corollaries of mitophagy based on western blotting and not imaging could be examined, such as ubiquitination of *Mfn1*.

4) The most important conclusion is that *Mul1* and Parkin both function downstream of PINK1. Parkin has been shown to be activated by the PINK1 substrate phospho-ubiquitin. Does phospho-ubiquitin activate *Mul1*?

5) PINK1 and p62 knockout mice are viable and fertile and whether they properly eliminate paternal mitochondrial upon fertilization could yield a rigorous corroboration of those conclusions.

Reviewer #3:

The manuscript by Rojansky and Chan investigates the role of mitophagy in the process of elimination of paternal mitochondria, upon oocyte fertilization and early embryogenesis. They show that Parkin and MUL1 synergize in triggering sperm mitochondria degradation, likely by ubiquitination of mitochondrial proteins. PINK1, *Fis1, Tbc1d15*, and p62 are also shown to be necessary, since silencing of any of these components of the mitophagy pathway causes impairment of sperm mitochondria degradation. The findings were somehow similar to those in MEF cells subjected to OXPHOS stress in acetoacetate medium. The main significance of the findings is in the reaffirmation of the role of mitophagy in maternal inheritance of mtDNA, which had been proposed and then negated in previous literature. The structure of the study is somehow unusual, because sperm and oocyte experiments are alternated with MEF experiments, which almost constitute two different stories. While the general mechanism of mitophagy may be conserved in oocytes and MEFs in OXPHOS medium, the specificity of the process for sperm mitochondria elimination is unexplained by this mechanism. On the other hand, some key information missing in this study could help clarifying the process. Overall, this is an interesting study, but mechanistic details are still lacking.

1) The rationale of why the ubiquitination process should only occur in the embryo and not in the sperm, before fertilization is essentially based on the fact that silencing in the embryo results in delayed elimination. However, it would be important to know if some components of the pathway are already tagging sperm mitochondria, even though the whole process is terminated after fertilization.

2) The levels of the key components of the mitochondrial ubiquitination pathway in sperm or in embryos before and after silencing are not shown. The former at least should be easy to investigate, in relation to the first point.

3) The MEF experiments are convincing, but do not illustrate novel mechanisms. It is unclear why they belong in this study on sperm mitochondria elimination.

4) The driver of the loss of function of sperm mitochondria hours after fertilization was not investigated. This is a fundamental question, under the premises that these mitochondria become targeted for mitophagy after fertilization. If some triggering components of the machinery were assembled prior to fertilization, in the sperm, the explanation would have to be searched in these cells.

5) If an unexplained loss of membrane potential were the trigger for mitophagy, then one would expect to see it occurring even if mitophagy is blocked by silencing of it components. This experiment was not performed. The opposite would suggest that depolarization could be the effect of a damage inflicted onto sperm mitochondria.

6) It should be determined if the loss of sperm mitochondria function could be triggered by insufficient maintenance by the embryo. For example, don't sperm mitochondria fuse with oocyte mitochondria? If not, why? Could the fusion machinery of the sperm mitochondria be insufficient?

---

## [Author Response]

*[…] One main difficulty that comes through the comments of all three reviewers is the challenge to assess the specificity of the mitophagy process you describe for paternal mitochondria. Are maternal mitochondria in zygotes also subjected to a similar process or is it specific to paternal mitochondria? This question may be addressed by using the same mutant mouse line but this time using female mutants crossed with wild-type males and while the number of maternal mitochondria may be large, you may be able to limit your quantitation by photoconverting Dedra in discrete areas of the zygote or by quantifying a fixed number of fields.*

We agree that the specificity of the mitophagy process is important to address. To examine this issue, we monitored the fate of mito-Dendra2 in the embryo at 36-84 hours post- fertilization depending on whether the fluorophore originated from the sperm or the egg. These new data are presented in Figure 1 and quantified in Figure 1—figure supplement 1 of the revised manuscript. When mito-Dendra2 arose from the male, the total fluorescence signal declines sharply at 72 and 84 hours, consistent with our earlier observations. In contrast, when mito- Dendra2 arose from the female, the total fluorescence signal shows no decline throughout the monitoring period. These results clearly indicate that the mitophagy process is specific for paternal mitochondria.

Related to this point, we also found that paternal mitochondria, unlike maternal mitochondria do not fuse with other mitochondria in the embryo (Figure 6—figure supplement 1). This segregation of paternal mitochondria is likely relevant to their eventual degradation.

*The use of glucose-free, acetoacetate medium also raised a number of questions. Not much is known about the effects of this medium on the biology of the cell beyond its action on OXPHOS. Could this medium have other effects that confound the interpretation of the results in MEFs? Are these culture conditions relevant to the more physiological conditions in the zygote? If the zygote has a more glycolytic metabolism, and the machinery identified here required an enhanced OXPHOS metabolism, how accurate can the identified machinery be? Clearly, some core elements such as Parkin and Mulan have been identified and are sufficient to modulate mitophagy in zygotes but these questions regarding the appropriateness and relevance of the culture medium must be addressed.*

This is an interesting issue – we did not intend to imply that OXPHOS-inducing media conditions bear a physiological relationship to early embryonic growth. We used the OXPHOS- inducing condition simply as an experimental tool to promote higher levels of mitophagy, whose molecular basis could be assessed. We then asked whether the molecules identified (MUL1, PARKIN, P62, FIS1, TBC1D15) are relevant to paternal mitochondrial degradation. The two conditions may in fact bear little direct relevance (in terms of metabolism) to each other. This issue is intriguing, because we may have identified mitophagy machinery that is relevant to mitochondrial degradation in a broad range of cell contexts.

In the revised manuscript, we have added the following text to the Discussion make this point clear:

"It is unclear whether MEFs cultured under OXPHOS conditions bear any physiological relation to the early embryo. Nevertheless, we find that the genetic requirements for removal of paternal mitochondria in the embryo mirror those of MEFs undergoing mitophagy in response to OXPHOS induction."

*This led us to raise the question of membrane potential. Was the machinery of mitophagy that you identified validated under normal membrane potential? In the last paragraph of the Results section you indicate that zygotes show a time-dependent loss of membrane potential, suggesting that this trigger is important to the autophagy of paternal mitochondria. From your data you do not demonstrate that the loss of membrane potential is instrumental here. It is possible that the loss of membrane potential is an epiphenomenon, unrelated to the autophagy of paternal mitochondria, since the machinery of mitophagy you report herein has not been identified in the context of loss of membrane potential. It would thus be helpful to at least validate your identified mitophagy machinery under an induced membrane potential defect. Formal assessment in the zygotes of the membrane potential in maternal mitochondria should also be provided.*

We agree – loss of membrane potential is an attractive trigger for mitophagy, but its functional role in paternal mitochondrial degradation cannot be definitively concluded from the available evidence. Given the known role of PARKIN in membrane-potential-dependent mitophagy, and the data showing loss of membrane potential in paternal mitochondria, it is reasonable to suggest that loss of membrane potential may play a role in triggering mitophagy in the early embryo. But it remains possible that it is an epiphenomenon. In the revised text, we state this point explicitly in the Discussion:

"Because we find that paternal mitochondria lose membrane potential shortly after entering the oocyte, it is tempting to speculate that this membrane depolarization may be the trigger for mitochondrial degradation. […] However, we do not have direct evidence that membrane depolarization has a functional role in paternal mitochondrial degradation."

In the cell culture assay, we found that CCCP-induced mitophagy is substantially reduced in MEFs lacking PARKIN and MUL1, but not cells lacking only one of these proteins. This result is presented in Figure 4—figure supplement 1.

Figure 6 shows that maternal mitochondria retain membrane potential, in contrast to paternal mitochondria.

*Your cellular and subcellular quantification methods were also questioned, and it would be important to better define how many elements were counted, how they were selected, and in how many cells from the embryos. Along this line it may be helpful to more clearly stress in the results that you counted all of the cells in each embryo and all paternal mitochondria in each cell. If, in some instances, this was not the case, then you must clarify how cells and/or mitochondria were selected and whether corrections were made for different number of cells per embryos of different ages. In addition, it was not clear what the level of detection is of Dendra-containing mitochondria in the microscope. Is the precision of your method and probe sufficient to confirm that one counted object equals one organelle? Also not clear from the manuscript was how you deal with detection of two mitochondria that are the fragmented/fissioned daughters of a parental organelle.*

All viable embryos from each experiment were imaged in their entirety. First, the top and bottom positions for each embryo were identified and used to capture z-stacks that recorded the entire embryo. Optical slices were taken at 1.1μm thickness, and z stacks were oversampled at 0.467 μm to ensure that all mitochondria were captured. Maximum intensity projections of these z-stacks were used for quantification. While it is true that embryos from the same litter can contain different numbers of cells, by utilizing maximum intensity z-projections of the full embryo, all mitochondria were counted regardless of slight variations in developmental stage.

The control and experimental embryo images were randomized and counted blind. Green fluorescent signals were counted manually. A mitochondrial object was defined as a green fluorescent structure with visibly distinct borders. In cases where two or more mitochondria could not be resolved definitely as separate organelles, they were counted as one mitochondrial object. As indicated in the representative images in Figure 5, the size of mitochondrial objects was within the range of 0.4 μm to 4 μm. Each maximum intensity z-projection was categorized as having either no mitochondria, less than five mitochondrial objects, or five or more mitochondrial objects.

These methodological details are now included in the Materials and methods section.

*Lastly, is it possible to photoconvert a single mitochondrion and follow its fate (sounds technically daunting but would be truly informative)?*

We performed photoconversion experiments with paternal and maternal mitochondria and found that they behave very differently. The photoactivated signal in maternal mitochondria diffuses over time and positive pixels decreases in intensity, indicating extensive fusion with adjacent mitochondria. In contrast, the photoactivated signal in paternal mitochondria does not change over time, indicating a lack of fusion activity. In the revised manuscript, these data are included in Figure 6—figure supplement 1.

*As you consider revising your manuscript, please also provide more detailed information about the statistical treatment of the data as requested by reviewer #1. Also, it would also be important to provide further detail and additional controls for the knockdown experiments to assure that the correct targets were indeed silenced and the effects do not reflect off-target effects, as discussed by reviewer #2.*

In the revised manuscript, we have included detailed information about the statistical analysis and have included an excel file with the relevant statistical details for the figures.

In Figure 4—figure supplement 1, we present the control experiments discussed by reviewer #2. These results indicate that the shRNA targets are effectively silenced. In addition, we show that mitophagy is robustly rescued by shRNAi-resistant constructs, thereby addressing the issue of off-target effects.

*Finally, we would like to provide our editorial position with respect to two more points. First, with respect to the novelty of the study, as questioned by reviewer #3 ("The MEF experiments are convincing, but do not illustrate novel mechanisms"), it is our position that the novelty of the study does not reside in the "novel mechanism" of mitophagy, but in the demonstration that mitophagy (even if it uses a known mechanism) is involved in the elimination of paternal mitochondria.*

*Second, with respect to some of the mechanistic gaps, primarily pertaining to how and when paternal mitochondrial are recognized (e.g. reviewer #3, points #1, 2, and 6), these are valid questions but are beyond the scope of this specific study. However, a valuable approach to at least provide support to the specificity of the mechanism for the paternal mitochondria would be to compare intra- vs. inter-species mitophagy. Indeed, the phenomenon of paternal mitochondrial incompatibility applies only to intra-specific matings in that in inter-specific matings the paternal mitochondria survive, at least in mice (Kaneda et al., PNAS 92:4542, 2015). Consequently, an inter-specific mating should essentially abrogate all the relevant phenomena found here in the intra-specific mating (e.g. the decay of Dendra signal), thereby strongly supporting your conclusions.*

We appreciate the editorial position concerning mechanism.

The suggested inter-specific experiment is very insightful. In the Kaneda et al. study, *M. spretus* males were mated to *M. m. domesticus* females to generate F1 hybrid embryos. PCR-based analysis showed that 25 of 45 neonates had detectable *spretus* mtDNA. Thus there is a variable amount of paternal transmission that occurs in this interspecific cross. As discussed in the Kaneda et al. paper, this result suggests that *domesticus* factors existing in the egg likely operate with reduced efficiency on the *spretus* mitochondria.

We were intrigued by this result and pursued it further. However, the only practical way to perform this experiment would be in the reverse direction of the Kaneda et al. study. We mated our *domesticus* males (containing mito-Dendra2) with commercially available *spretus* females, which we primed with hormone. Unfortunately, when we flushed the oviducts of such females, we found only a few abnormal, nonviable embryos.

We examined further the literature on interspecific mouse matings and found that the direction of the cross is crucial: *domesticus* males do not mate successfully with *spretus* females. For example, Zechner et al. (Nature Genetics 12: 398-403, 1996) states that "While the (*mus* x *spr*) F_1_ offspring are readily obtained, the reciprocal (*spr* x *mus*) F_1_ are notoriously difficult to breed." [Note: In this notation, the female parent is listed first.] A review (Dejager et al. Trends in Genetics 25: 234-241, 2009) on *spretus* interspecific genetics states: "Although viable F_1_ hybrids can be easily produced by mating a *M, spretus* male with a female of most classical inbred strains, the reciprocal crosses are almost invariably sterile."

Moreover, F_1_ males from *mus* x *spr* crosses are sterile due to lack of sperm cells. As a result, it is not possible, through mating, to construct a *spretus* male hybrid with the mito-Dendra2 allele.